# Geospatial Monitoring of Land Surface Temperature Effects on Vegetation Dynamics in the Southeastern Region of Bangladesh from 2001 to 2016

**Shahidul Islam** [1,2,3] and **Mingguo Ma** [1,2,3,*]

1   Chongqing Engineering Research Center for Remote Sensing Big Data Application, School of Geographical Sciences, Southwest University, Chongqing 400715, China; shahid_ges@yahoo.com
2   Research Base of Karst Eco-environments at Nanchuan in Chongqing, Ministry of Nature Resources, School of Geographical Sciences, Southwest University, Chongqing 400715, China
3   Chongqing Jinfo Mountain Field Scientific Observation and Research Station for Kaster Ecosystem, School of Geographical Sciences, Southwest University, Chongqing 400715, China
*   Correspondence: mmg@swu.edu.cn; Tel.: +86-23-6825-3912

**Abstract:** Land surface temperature (LST) can significantly alter seasonal vegetation phenology which in turn affects the global and regional energy balance. These are the most important parameters of surface–atmosphere interactions and climate change. Methods for retrieving LSTs from satellite remote-sensing data are beneficial for modeling hydrological, ecological, agricultural and meteorological processes on the Earth's surface. This paper assesses the geospatial patterns of LST using correlations of the seasonally integrated normalized difference vegetation index (SINDVI) in the southeastern region of Bangladesh from 2001 to 2016. Moderate Resolution Imaging Spectroradiometer (MODIS) time series datasets for LST and SINDVI were used for estimations in the study. From 2001 to 2016, the MODIS-based land surface temperature in the southeastern region of Bangladesh was found to have gently increased by 0.2 °C ($R^2$ = 0.030), while the seasonally integrated normalized difference vegetation index also increased by 0.43 ($R^2$ = 0.268). The interannual average LSTs mostly increased across the study areas, except in some coastal plain and tidal floodplain areas of the study. However, the SINDVI increased in the floodplain and coastal plain regions, except for in hilly areas. Physiographically, the study area is a combination of low lying alluvial floodplains, river basin wetlands, tidal floodplains, tertiary hills, terraced lands and coastal plains in nature. The hilly areas are mostly covered by dense forests, with the exception of agricultural areas. The impacts of increased LSTs were inversely correlated for the hilly areas and areas with forest coverage; LSTs were conversely correlated for the floodplain region, and tree cover outside of the forest and agricultural crops. This study will be very helpful for the protection and restoration of the natural environment.

**Keywords:** LST; seasonally integrated normalized difference vegetation index (SINDVI); correlation analysis; MODIS; remote sensing

## 1. Introduction

As one of the most important parameters of surface–atmosphere interactions, land surface temperature (LST) plays a crucial role in modeling hydrological, ecological, agricultural and meteorological processes on the Earth's surface [1,2]. The land surface temperature is increased by anthropogenic heat discharge due to energy consumption and associated decreases in vegetation and water-pervious surfaces which reduce the surface temperature through evapotranspiration [3]. Thermal remote-sensing data can be used to retrieve the spatially distributed LSTs by measuring the upward long-wave radiation from the land surface under clear-sky conditions. The thermal signal

acquired by a remote sensor at the top of the atmosphere (TOA) is influenced by surface parameters, e.g., temperature and land surface emissivity [4]. In addition to providing measurements of radiant surface temperature, remote-sensing instruments also collect measurements of reflecting energy in the red and near-infrared portions of the electromagnetic spectrum that can be used to quantify the extent of changing conditions in vegetation [5].

LST is one of the key factors in the physics of land surface processes, combining surface–atmosphere interactions and the energy fluxes between the atmosphere and ground. The lowest LSTs are usually found in dense vegetative areas, although this can differ with the times, places and types of vegetation distributions [6]. Weng et al. (2004) [7] found that the vegetation fraction has a slightly stronger negative correlation with LST. Yue et al. (2007) [5] presented that the mean LST and normalized difference vegetation index (NDVI) values associated with different land-use types are significantly different. Joshi and Bhatt (2012) stated that areas with vegetation and water bodies have lower temperatures compared to built-up areas [8]. Sun and Kafatos (2007) [9] found that the correlation between LST and NDVI is positive during the winter and negative during warm seasons.

Developing countries, as well as the rest of the underdeveloped world are the most often impacted victims of climate change. As a developing country, Bangladesh is one of the most vulnerable countries in the world due to climate change [10]. The study area, the southeastern region of the country, is one of the most affected and visibly impacted areas of them. As a crucial environmental process and an important indicator of climate change, the land surface temperatures in this area are increasing gradually. There have been a number of research studies done on climate change based on the air temperature and vegetation dynamics in Bangladesh, while land surface temperature (LST) was limited in use. Ahmad et al. [11] reported an increase of 0.5 °C in temperature over Bangladesh over the past 100 years. Mondal and Wasimi [12] analyzed the temperatures and rainfall of the Ganges Delta in Bangladesh and found an increasing trend of 0.5 °C and 1.1 °C per century in day-time maximum and night-time minimum as well. Based on regional trends in temperature and rainfall, they concluded that water scarcity in the dry season might increase and the critical period could become more critical in the future. The SAARC Meteorological Research Centre (SMRC) [13] studied surface climatological data and showed an increasing trend of mean, maximum and minimum temperatures in some seasons, and decreasing trends in some others. Overall, the trend of the annual mean maximum temperature showed a significant increase over the period of 1961–90. Rahman and Alam [14] found that the temperature is generally increasing during the June–August period. Average maximum and minimum temperatures showed an increasing trend of 5 °C and 3 °C per century, respectively. On the other hand, average maximum and minimum temperatures of the December–February period showed a decreasing and increasing trend of 0.1 °C and 1.6 °C per century, respectively [15].

The southeastern region of Bangladesh is a densely populated region where vegetation and trees outside of forests play an important role in the national economy and carbon sequestration. The natural vegetation of this region consists of tropical moist deciduous and semievergreen forests, mangroves, and fresh water wetlands [16]. The eastern forests belts of the study area, at the border with Myanmar, are related to the Indo-Burma biodiversity hotspot, one of the few globally significant areas with high species diversity and endemism [17]. The total national tree canopy cover area including trees outside forests makes up 54% of the total canopy cover [18]. Potapov et al. 2017 [18] found that the total tree canopy cover increased 4.3% during the 2000–2014 time interval; forests decreased their tree cover area by 83,600 ha, and the trees outside of forests (including tree plantations, village woodlots, and agro-forestry) increased their canopy area by 219,300 ha. Bangladesh exhibits a national tree cover dynamic where the net change is rather small, but gross dynamics are significant and variable by forest type. Despite the overall gain in tree cover, the results revealed the ongoing clearing of natural forests, especially within the southeastern hill tracts. As a climatic victim area, the vegetation dynamics and distributions of the southeastern region of Bangladesh have a crucial relationship with LST. Based on these criteria, we have not identified any studies similar to the research presented here.

Therefore, the main purpose of the study was to monitor the geospatial relationship of LST and vegetation dynamicsas well the seasonally integrated normalized difference vegetation index (SINDVI) in the southeastern region of Bangladesh over the time period 2001–2016. SINDVI is defined as the sum of NDVI values for each pixel and the all-time intervals of the maximum value composites (MVCs) [19] for which the NDVI exceeds a critical value (commonly NDVI > 0.1) [20]. The working hypothesis tested in this study is that the spatial distribution of SINDVI at the study areas depends on LST, and that the LST range in each image is related to the phenological cycle. From a methodological perspective, this study relies on the potential of remotely sensed data and, more specifically, Moderate Resolution Imaging Spectroradiometer (MODIS) time-series products for LST and SINDVI estimations.

## 2. Materials and Methods

### 2.1. Study Area

The study area, the southeastern region of Bangladesh (Figure 1a) known as Chattogram Division (former Chittagong Division) [21], borders the Bay of Bengal in the south, India and Myanmar in the east and the locally largest river, the Meghna River (based on water discharged), in the north and west. Geographically, it is the largest administrative division of Bangladesh out of eight, with 33,904 sq. km (13,090.41 sq. mi) land surface [22] and a population of 29.15 million consisting of a density of 884 persons per sq. km [23]. The geographical location of the area is 22°40′ N to 24°10′ N and 90°45′ E to 92°40′ E. The northern and western portions are listed as low-lying alluvial floodplains of the Meghna River that are less than 10 m above sea level, comprising 37.6% of the region. However, the remaining southern and eastern portion, where the elevation exceeds 200 m, comprises 62.4% of the area, mostly with a south–north distributed hilly nature [23]. The eastern portion hilly regions are mostly covered by dense forest with a variety wild life.

Physiographically, the study area is the combination of 8 different types of lands according to the Bangladesh Agricultural Research Council (BARC) of the government (Figure 1b). The major portions of this region are occupied by the northeastern hills in a tertiary formation of lower Tibetan plateau. The vast areas of plain lands are mainly alluvial river floodplains, estuarine floodplains, wetlands, tidal floodplains and coastal plains in nature. Additionally, a small portion of terraced land was also found in the study areas.

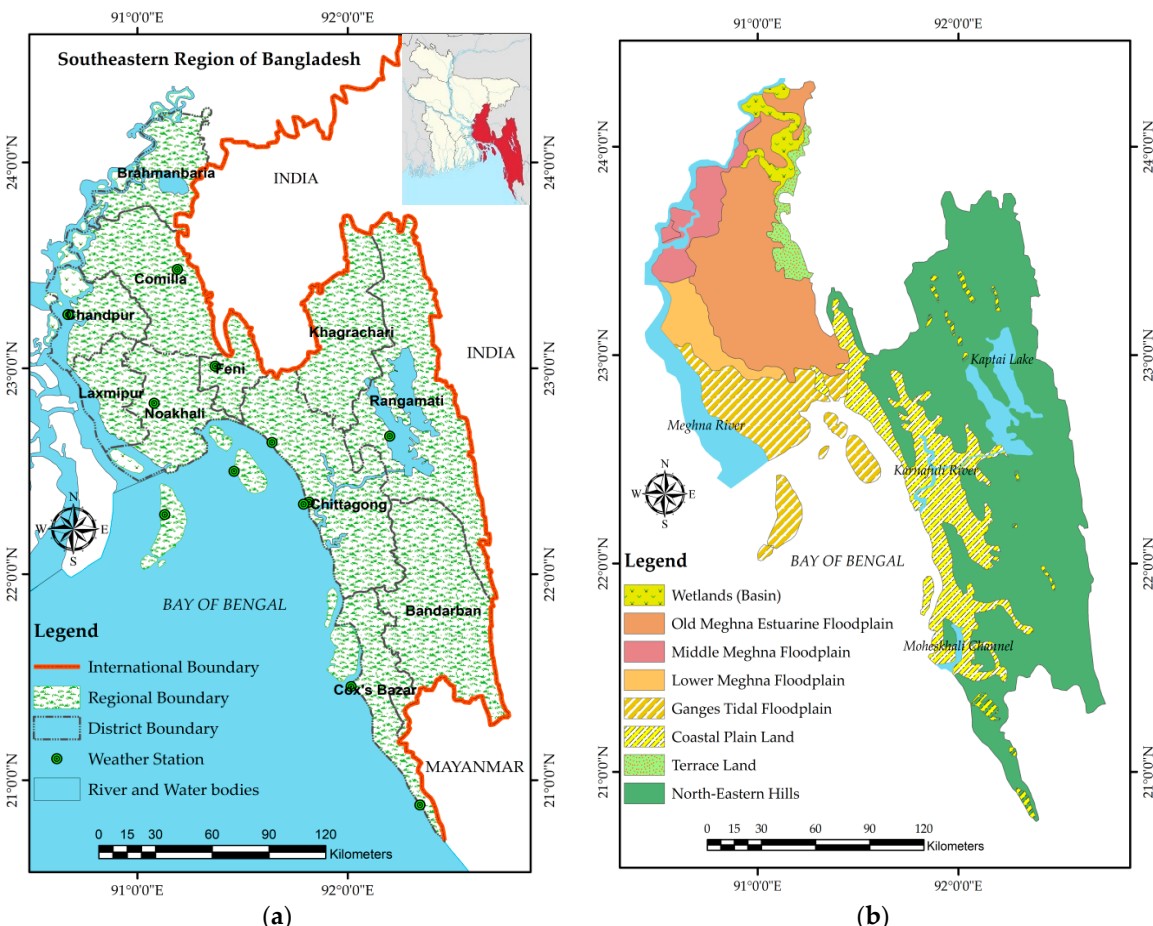

**Figure 1.** (**a**) Location of the study area and (**b**) physiographic character ofthe southeastern region of Bangladesh (Source: Banglapedia and BARC, 2010).

## 2.2. Datasets

To monitor and quantify the surface dynamics of heterogeneous landscapes, high temporal and multiresolution synthetic time series of the LST and NDVI datasets were generated from MODIS. The datasets were evaluated regionally according to the study area in the southeastern region of Bangladesh to ensure that its accuracy can meet the needs of an analysis of spatiotemporal climate change [24]. The MODIS LST and NDVI datasets were obtained from the Land Processes Distributed Active Archive Center (LP DAAC) version v006 managed by the National Aeronautics and Space Administration (NASA) Earth ScienceData and Information System Project with a geographic latitude/longitude projection at 1 kilometer.

### 2.2.1. Land Surface Temperature (LST)

The MODIS Terra eight-day LST composition period products (MOD11A2, C6) were chosen because two of such periods are the exact ground track repeat period of the terra platform [25]. The datasets were used over eight days with averaged LST products on clear sky days and clear sky nights of 1 km spatial resolution between January 2001 and December 2016. For calculating the trends of the LST of the study area, there were 736 MODIS LST dataset-based images that were analyzed during the period of study. The yearly composites of LST were derived from the average of 46 MODIS LST products from the eight-day compositing period. A simple average method is used in the current algorithm on each pixel and the all-time intervals of maximum value composites (MVCs) [19,26].

The impact of the number of clear-sky observations per pixel of the 2016 composites period of LST datasets were analyzed as clear-sky day and clear-sky night (Figure 2a,b). Overall, approximately 33 clear-sky day and 34 clear-sky night observations were identified out of 46 composites of LST in 2016.

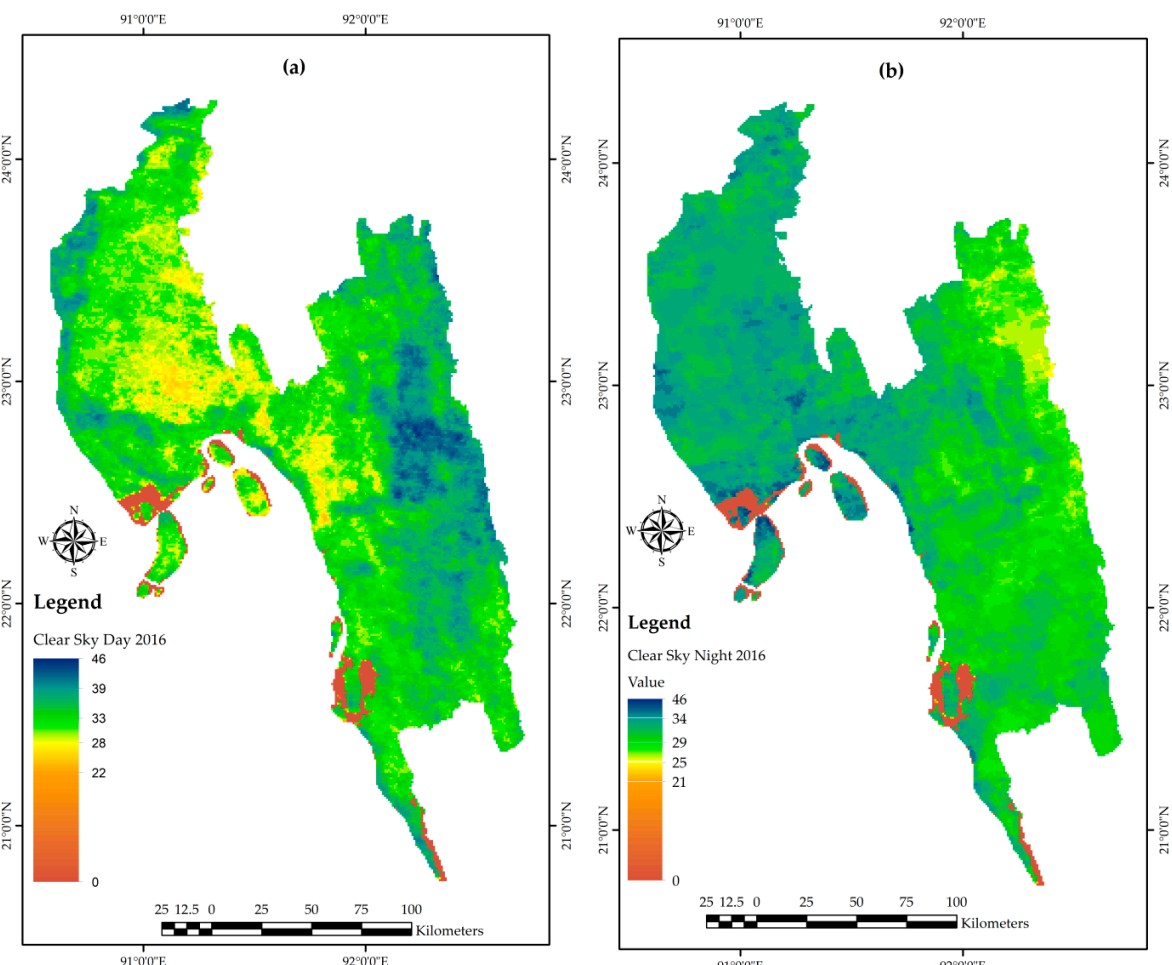

**Figure 2.** The total number of observations per pixel of (**a**) a clear sky day and (**b**) a clear sky night for the final composites of land surface temperature (LST) in 2016.

2.2.2. Normalized Difference Vegetation Index (NDVI)

Datasets between January 2001 and December 2016 of MODIS Terra monthly NDVI products (MOD13C2, C6) based on cloud-free spatial composites of 16 days of 1 km spatial resolution were used for NDVI values identification. The SINDVI was used to characterize the study areas of vegetation states and processes. Twice-monthly NDVI composite data sets were integrated over each growing season to derive maps of SINDVI for each year. This was achieved by summing mean NDVI values of $3 \times 3$ pixel blocks (to minimize misregistration effects), for each of 16 day periods, from 1 January to 31 December of each year [27]. The SINDVI was described by each pixel's NDVI sum values when the NDVI exceeds a threshold value (commonly defined as NDVI > 0.1) [20]. In this study, only the grid cells with greater than 0.1 were used in order to eliminate the influence of bare and sparsely vegetated regions [28,29] as well as to determine the growing season [30]. There were 368 of MODIS NDVI dataset-based images that were analyzed over the 16 years of study period to calculate the trends of the SINDVI in the study area, as 23 images were covered each year.

### 2.3. Statistical Analysis

The yearly maximum, minimum and average values of land surface temperatures were estimated for time series analysis by averaging the interannual data. A suitable correlation analysis was built up between average interannual data of the LST and SINDVI.

#### 2.3.1. Time-Series Analysis

To estimate the variation trends of the spatial patterns of the MODIS LST and SINDVI, the annual change tendencies were quantified at a regional scale. Monthly time series were processed to form yearly time series; for each pixel, the time series has a length $n = 16$, spanning from 2001 to 2016. Change rate and change range were used to estimate regional tendency based on the linear regression and appeared to be unaffected by possible constant biases present in the data [20,27,31–33]. Therefore, the slope and range were used to assess the interannual variations of LST and SINDVI [20,31,33]. The slope represents the change rate of each pixel calculated by the ordinary least-squares estimation via linear regression from 2001 to 2016. Ordinary least squares (OLS) methods in the linear regression were applied as:

$$Slope = \frac{n \times \sum_{i=1}^{n}(i \times Ai) - \sum_{i=1}^{n} i \times \sum_{i=1}^{n} Ai}{n \times \sum_{i=1}^{n} i^2 - \left(\sum_{i=1}^{n} i\right)^2}. \tag{1}$$

where $n$ is the length of the time series that was studied; $i$ is the number of years, $i = 1,2, \ldots , n$, in this paper, $n = 16$; and $Ai$ means the LST or SINDVI in the $i$th year. When the slope values near 0, it means there are no significant changes in the trend. Slope $> 0$ indicates an increasing trend, while slope $< 0$ means a decreasing trend [34].

Afterward, the ranges of variations (the total change between 2001 and 2016) in the LST and SINDVI cover were calculated:

$$Range = Slope \times (n - 1) \tag{2}$$

where $n$ is the length of the time series as $n = 1, 2, \ldots , 16$.

The conventional non-parametric Mann–Kendall test (M–K test) has been extensively used to assess the significance of monotonic trends in the LST and SINDVI time series [35–37].

#### 2.3.2. Correlation and Partial Correlation Analysis

To determine the correlations for each pixel, Pearson's correlation coefficients were calculated between the LST and SINDVI [31]. From January 2001 to December 2016, the yearly composite values of LST (value > 1.96K) and the yearly SINDVI (NDVI > 0.1) of each pixel and their annual average values in total time series (16 years) were used to calculate the correlation.

$$r_{XY} = \frac{\sum_{i=1}^{n}(X_i - \overline{X})(Y_i - \overline{Y})}{\sqrt{\sum_{i=1}^{n}(X_i - \overline{X})^2}\sqrt{\sum_{i=1}^{n}(Y_i - \overline{Y})^2}} \tag{3}$$

where, $r$ is the correlation coefficient between $X$ and $Y$; $Xi$ and $Yi$ are the values of the $i$th year and $X$ and $Y$ are the annual average values of the time series; $i = 1,2, \ldots , n$, in this paper, $n = 16$.

In addition, for the correlation calculation, Student's $t$ test was applied to assess statistical significance and a table of critical values of Pearson correlations was used for significance distribution [31].

### 2.4. Reconstructed NDVI Data

Timeseries of NDVI datasets still possessed significant residual effects and noise levels. A simple but more efficient method, the mean-value iteration filter (MVI) was applied to reduce the noise and fill the gap of the monthly spatial composites of 16 days of NDVI time series data [38]. This method has been proven to be more effective in reconstructing temporal and spatial NDVI time series in comparison to other methods (best index slope extraction (BISE) algorithm, the modified BISE algorithm, and a

Fast Fourier Transform algorithm), commonly used as a reconstruction approach [38–41]. The choice of the threshold value is very important in the newly developed MVI method. Most points would be retained or adjusted if threshold value were too high or low as well [38].

For each pixel, the multiyear average NDVI (ANDVI) was calculated to replace the no-data pixels for the same month, estimated as follows:

$$\Delta_i = [NDVI_i - \frac{(NDVI_{i-1} + NDVI_{i+1})}{2}] \tag{4}$$

where $i$ indicates $i$th observation of the monthly $NDVI$ ($i$ varies from 1 to 180 for the 16 years). When $\Delta_i$. is greater than a threshold value, $NDVI_i$. is replaced by $(NDVI_{i-1} + NDVI_{i+1})/2$. The threshold value ($\Delta_T$) can be set as a small percentage of the multiyear ANDVI for each pixel. Finally, the iteration ends when the entire $\Delta_i$. are less than $\Delta_T$.

### 2.5. Deseasoned Anomalies

The seasonal cycle was removed by subtracting the average value for the same month over years as follows [42,43]:

$$\Delta_{\propto(yy.mm)} = \propto (yy.mm) - \overline{\propto}(mm). \tag{5}$$

where $\Delta_{\propto(yy.mm)}$. is the deseasoned LST for year $yy$ and month $mm$, $\alpha(yy.\ mm)$ is the monthly LST in year $yy$ and month $mm$, and $\overline{\propto}(mm)$ is the average LST over 16 years for month $mm$.

### 2.6. Research Design

The geospatial relationship monitoring of land surface temperature effects on vegetation dynamics in the southeastern region of Bangladesh from 2001 to 2016 was the principal objective of this research. To fulfil the objective, this research tried to determine the interannual average and existence trends of LST and SINDVI values (2001–2016) at the study area. Their correlations were also evaluated and explored for the reasons of the variations in relation to each other. Meanwhile, depending on these results, the study made some specific discussions of a few extreme impacts and the overall geospatial relations between LST and SINDVI at the study area. The datasets of the study were processed and analyzed as depicted in the following flowchart (Figure 3):

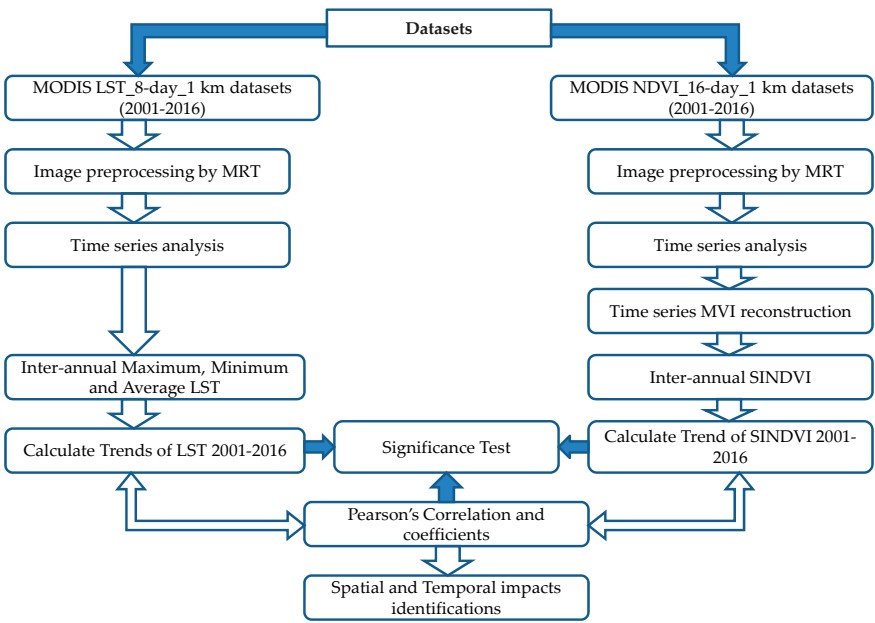

**Figure 3.** Data processing and analysis flowchart.

For each pixel, the slope over the past 16 years (2001–2016) was calculated for the LST and SINDVI by using the slope equation (Equation (1)). Once the slopes were identified, the range of their changes in this time period was estimated by Equation (2). A greater range indicates a more dramatic variation. The final ranges of these two variables were correlated by Pearson's correlation (Equation (3)). In addition, the correlation coefficients were analyzed using the SPSS 11.0 software (SPSS Inc, Chicago, IL, USA) as described in the equations to explore how the variation of the SINDVI can explain the changing LSTs. The absolute value of correlation, namely, the correlation between LST and SINDVI was compared for each pixel.

## 3. Results

### 3.1. Variation Trends of LST and Seasonally Integrated Normalized Difference Vegetation Index (SINDVI)

The interannual variation trends of LST and SINDVI over the time period 2001–2016 in the southeastern region of Bangladesh are shown in Figure 4a,b respectively. The overall changes in LST over the 16-year time period and trends of increased (red) and decreased (green) temperatures were expressed as degrees Celsius (°C). The greenness rate of change (GRC) was defined here as the slope of the least-squares line fitting the interannual variability of the SINDVI values over the time period [20]. While changing trends in vegetation ranges of the SINDVI increased (green) and decreased (red), they were expressed as their indexed values.

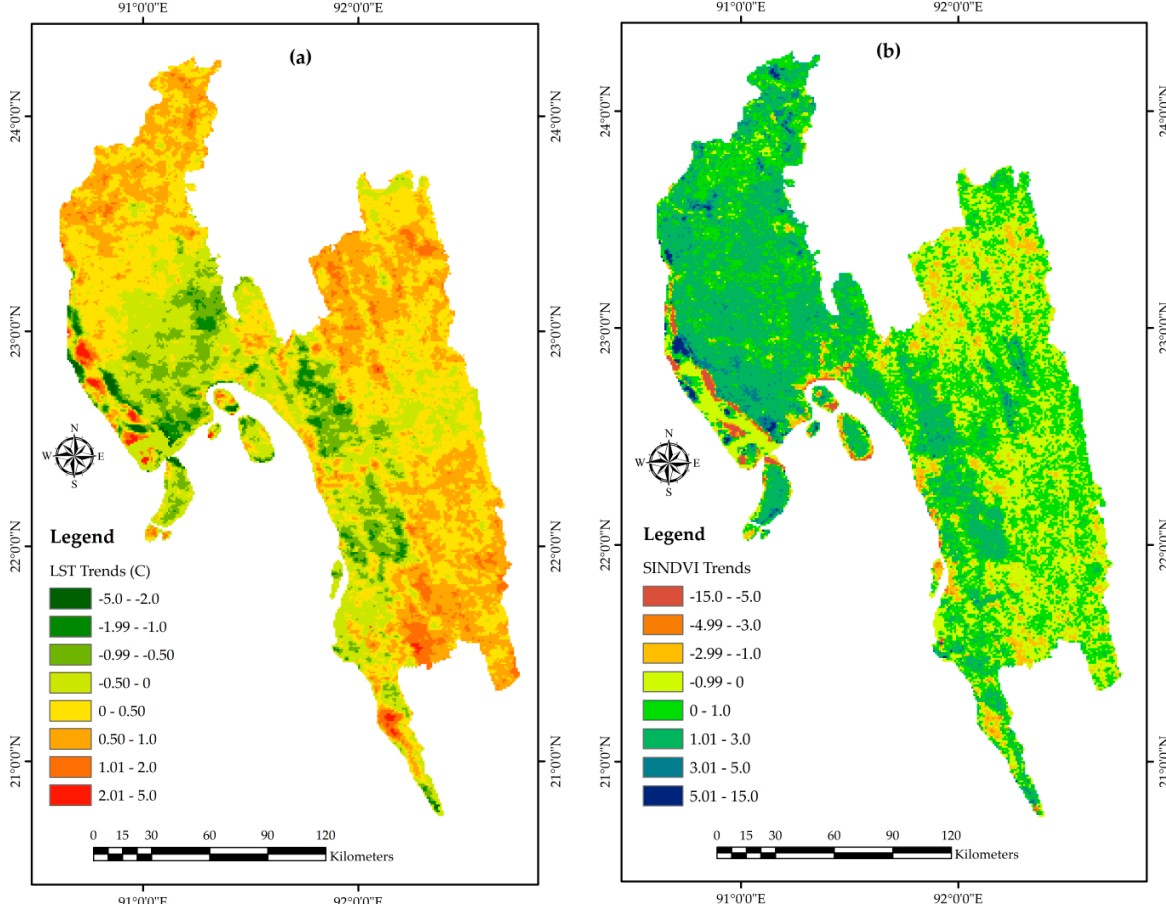

**Figure 4.** Variation trends of (**a**) LST, and (**b**) seasonally integrated normalized difference vegetation index (SINDVI) during the period of 2001–2016. Regional boundaries were modified from a shape file using ArcGIS 10.2 (Chongqing Engineering Research Center for Remote Sensing Big Data Application, Southwest University, China).

LSTs in the southeastern region of Bangladesh were found to be mostly increasing except in some specific regions of the study area. Overall, approximately 63% of the total study area shows an increase in LST during the period from 2001 to 2016 (Figure 4a). A range of highly increased LSTs were identified in the northwestern active floodplain region; in the river basin and wetlands areas of Brahmanbaria; and in Chandpur and the upper portion of the Comilla districts as well. At the same time, the hilly and mountain areas of Khagrachari, Bandarban, and some other portions of the Cox'sBazar and Rangamati districts were also marked as having increased LSTs. The highest temperature increases (2.01–5.0 °C/16 years) were recorded at Teknaf and Ukhia of Cox'sBazar and Faridganj of Chandpur in the study area. The changing trends of LSTs were mostly stable at the southern coastal belt areas of Chattogram and Cox's Bazar districts and some islands in the Bay of Bengal, as well as in the foothill and valley regions of the study. On the other hand, in the tidal floodplain of Noakhali district, the old Meghna estuarine floodplains of Comilla district, and coastal plain lands of Chattogram and Feni districts, temperatures were identified with trends of decreasing ranges (LST < −1 °C/16years). Overall, during the period of study (2001–2016), an approximately 0.2 °C increase in temperature has occurred in the study area.

The interannual existence trends of the SINDVI have also generally increased in the study areas and have demonstrated greening trends all over the region. Overall, 63.69% of grid cells were identified as having increased ranges of SINDVI in the years 2001–2016 (Figure 4b). The interannual increasing and decreasing trends of vegetation cover have classified the region as having a flat and uneven topography. The area of floodplains, wetlands, estuarine plains, tidal plains, and coastal plains have shown an increasing dynamism of vegetation (SINDVI > 0.1). On the other hand, the annual index of vegetation cover was in the trends of decreasing at the northeastern hill region as well as all over the hilly and mountain areas (SINDVI < −0.1). The surrounding areas of wetlands, river bank areas of middle Meghna floodplain and lower portion of tidal floodplain were identified as the highest increasing range (5.0–15.0/16 years) of SINDVI. The floodplain region of Bangladesh is characterized by fertile soil with varieties of agriculture and rural housing [15,18]. Remarkably decreased SINDVI trends (5.0–15.0) were mainly identified in the northern and southern portions of hilly areas of the Khagrachari and Bandarban districts. This region is mostly covered by evergreen and mixed reserve forests as one of the important forest regions of the country [18].

The application of M–K test statistics resulted in the identification of trend direction in the change of LST and SINDVI at the southeastern region of Bangladesh from 2001 to 2016. A negative trend indicates the decline of temperature or vegetation cover and a positive trend indicates the rise or increase over the years [35–37]. As each monitoring reflects the LST and SINDVI dynamics of the surrounding area, each trend value gives an idea about the temperature or vegetation fluctuations of the study area over the time period.

The Mann–Kendall (M–K) changing parameters (UB and UF) curve line shows a statistically significant increasing trend of LST from 2002 to 2006 and 2007 to 2009, while a statistically significant sharp increasing was identified from 2012 to 2016. The LSTs were significantly decreasing in the study area only from 2009 to 2012 (Figure 5a). On the other hand, the SINDVI was statistically significantly increasing between the years 2001–2003, 2004–2007, and sharply increasing during 2010–2016. Statistically significant sharp decreases were only identified between 2003–2004 and 2007–2010 (Figure 5b).

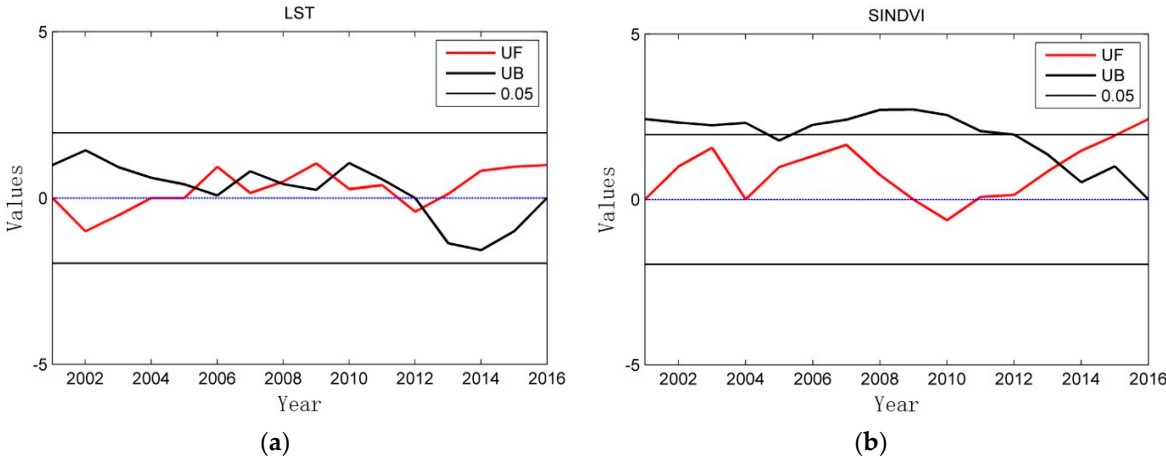

**Figure 5.** The significant trend of annual (**a**) LST, and (**b**) SINDVI at the southeastern region of Bangladesh from 2001 to 2016 (Note: U*B* = −U*F*; Significance tested at α = 0.05). The figure for the significance test of the trend of LST and SINDVI has been drawn by a non-parametric M–K test.

### 3.2. Interannual Variability

The interannual variability of LST and SINDVI were the summed monthly mean values of each grid cells for the respective year's LST and NDVI. To extract the interannual variability, a simple average method was applied in considering the interval of maximum value composites (MVCs). From 2001 to 2016, annual average LSTs increased 0.2 °C ($R^2$ = 0.030). The highest yearly average LST (301.2K) was 2014 and lowest anomalies (299.9K) was in 2012 (Figure 6a). High increases were observed from 2012 to 2014, of approximately 1.3 °C/year, and the highest decreases of 0.9 °C/year were observed from 2014 to 2016.

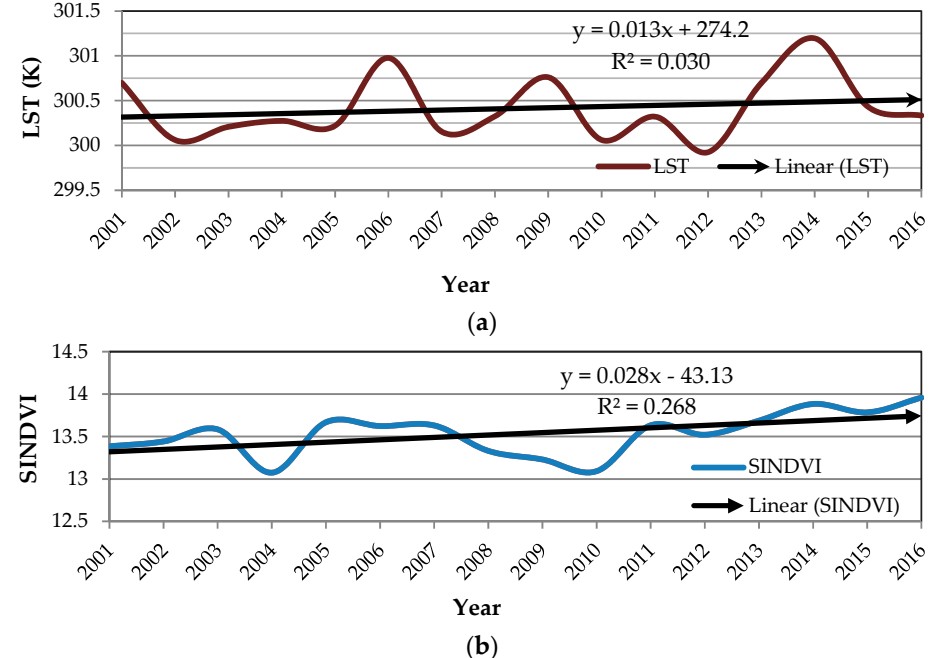

**Figure 6.** Interannual average (**a**) LST, and (**b**) SINDVI during 2001–2016 in the southeastern region of Bangladesh with linear variation trends.

The annual average SINDVI increased steadily for the entirety of 2001 to 2016 (Figure 6b). The indexed values of SINDVI were highest (14.0) in 2016 and lowest (13.1) in 2004 and 2010. The SINDVIs had increasing trends in 2004–2007 and 2010–2016. The indexes rapidly increased by 0.9

indexed values for the 2010 to 2016 period of study. Overall in the period of study, the inter-annual average SINDVI was increased by 0.43 ($R^2$ = 0.268) indexed values at the study areas.

### 3.3. Correlation with LST and SINDVI

To quantify the spatial correlation coefficients of the study area in the southeastern region of Bangladesh, the correlation coefficients of all grid cells between LST and SINDVI were calculated (Figure 7a). The Pearson's correlation coefficients were used to determine the correlations on each pixel by the annual average and inter annual time series average values of LST and SINDVI for 2001 to 2016 (Equation (3)). Approximately 58.26% of the total grid cells were found to be in a negative correlation and had an average correlation value of 0.049 ($R^2$ = 0.049, $p < 0.5$) in the study area. The LST and SINDVI were inversely correlated mostly at the tidal floodplains and southeastern hilly regions of the study. The vast areas of plain lands as well as river basin floodplain, wetlands, estuarine floodplains and foothill areas were mostly conversely correlated. Inverse correlation coefficients were mainly identified in the hill tracts region of Chattogram, Cox'sBazar, Khagrachari, Rangamati and Bandarban districts as a whole (SINDVI > −0.4). Some portion of the Ganges tidal plains of Noakhali district and coastal plain lands of Chattogram and Feni districts were also identified as having inverse correlations (>−0.2). Converse correlation coefficients were strongest (>0.7) in the middle Meghna floodplain region, river basin and wetlands areas of Brahmanbaria district. A moderately converse correlation (>0.3) was identified at the lower Meghna floodplain region of Chandpur district and old Meghna estuarine floodplain of Comilla district. Therefore, reserve forest and green belts were the areas affected by increased surface temperatures by the years 2001–2016 at the study area; however, the tree covers excluding forest received positive impacts.

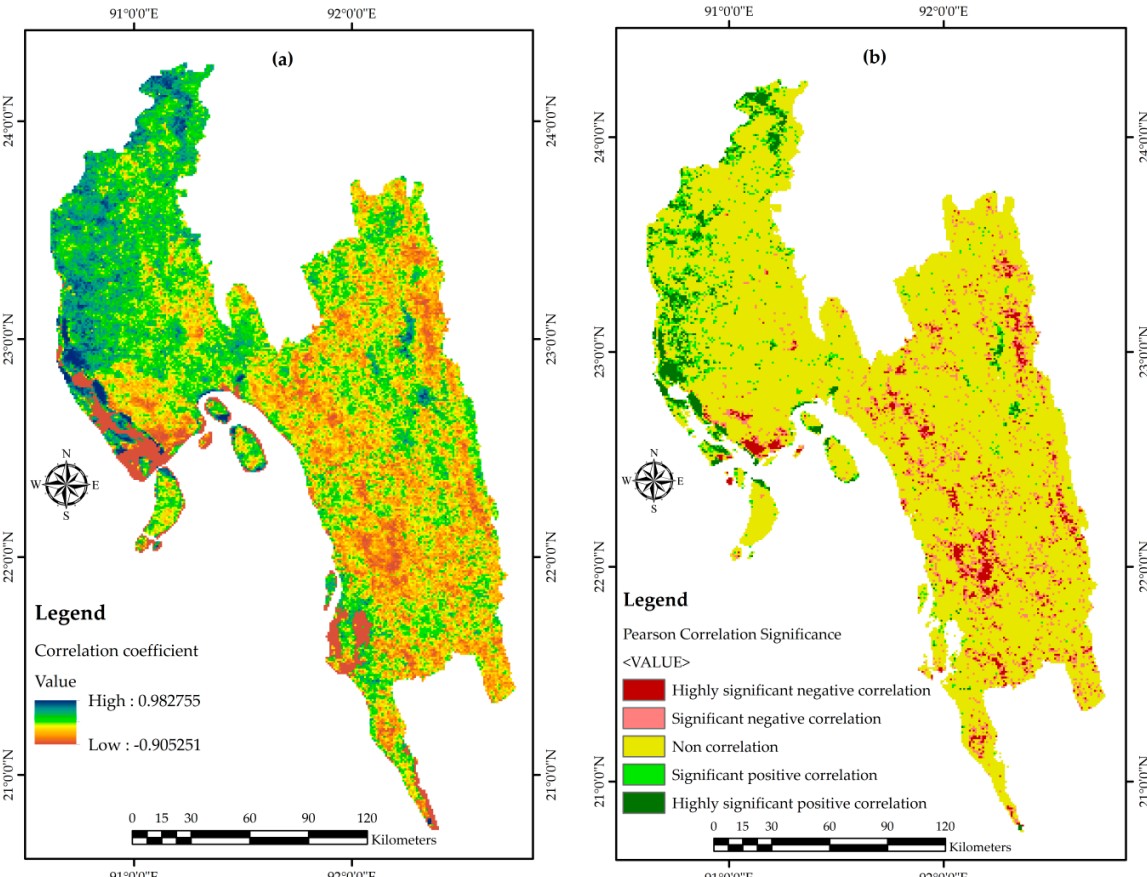

**Figure 7.** The spatial patterns of LST and SINDVI values' (**a**) Pearson correlation coefficient, and (**b**) Pearson correlation significance testing by critical values.

Pearson's correlation significance testing by critical values shows that the significant negative correlations were mostly distributed in the southeastern hilly areas and significant positive correlations were mostly in the flood plain areas as well as in flat topographic regions, except for the tidal floodplain of the study areas (Figure 7b). Highly negative correlations were mostly identified at the urban and built-up areas of the hilly region and the transitional land areas of tidal floodplain. Highly positive correlations were mostly found in the wetlands and the Meghna River floodplains, as well as the active deltaic region of the study.

## 4. Discussion

This study has demonstrated that different physiographic land types and their surface characteristics are highly related to LSTs and will affect the general pattern of the vegetation dynamics. Different types of land surfaces have been shown to vary greatly in their effects on LST and SINDVI. Both inverse and converse relationships between LST and vegetation abundance were observed in the study area as a whole. The time-series analysis of the interannual average LSTs demonstrated the increasing trends of temperatures all over the study area at the southeastern region of Bangladesh, except some small portions of foothills and coastal plain lands (Figure 4). The increasing rates of LSTs were considerably high at the hill top areas, all over the wetlands, and in some river basin areas of the study. Many more studies have shown that LSTs are inversely related with NDVI [6,7,9]. However, this study found the relationships between LST and SINDVI were fully or partially dependent on geospatial locations, physiographic land criteria and vegetation indexes as well. The interannual average LSTs and SINDVI were only related conversely in the annual temperature decreased areas of the study. Furthermore, both the increasing and decreasing trends of SINDVI were identified at the locations of increased LST areas. Overall, the increased LSTs at the study areas were positively impacted by vegetation cover out of the forest, while there were negative or inverse impacts on forest coverage.

### 4.1. Significant Areas of Extreme Impact

The cumulative ranges of the LST and SINDVI variation trends (Figure 4) have found some specific point areas of the study that changed in extreme ways during the period 2001–2016. There were six points identified with such variations of LST and SINDVI. These points of extreme impact were selected by their extremely increased (>4 °C) and decreased (>−4 °C) temperatures, extremely increased (>10.0) and decreased (>−10.0) SINDVI, and no changes of either elements in any areas of the study for their 16 years cumulative value of trends. From the inner areas of the identified areas with extremely impacted variation trends for LST and SINDVI, a single and the same grid cell was taken into consideration. The interannual average values for the grid cells from both trends were analyzed (Figure 8).

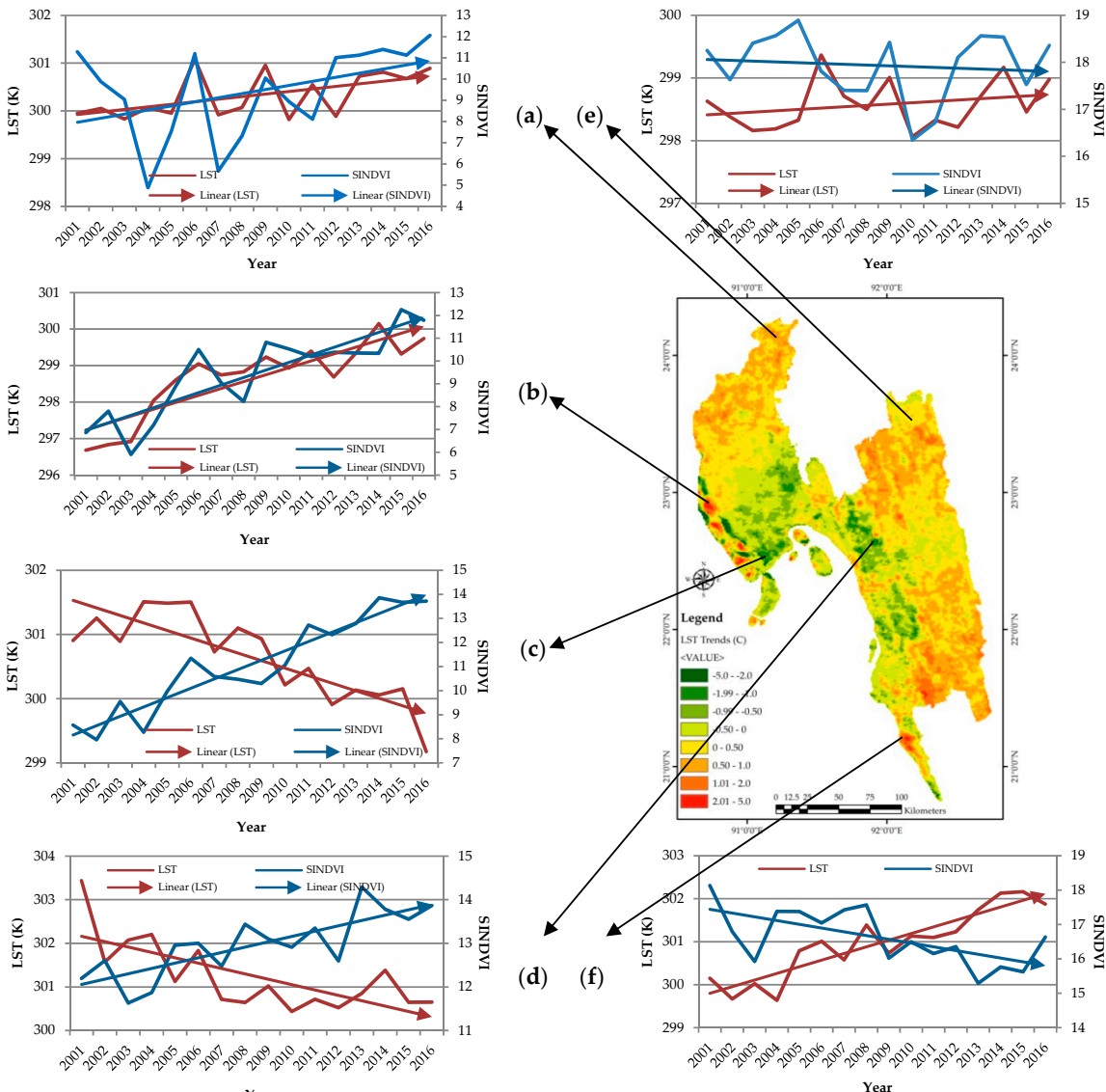

**Figure 8.** The interannual average LST and SINDVI of the extremely impacted grid cells area (0.008833 × 0.008833) for six different regions of the study area. The map in the main panel inserted represents the range of LST from 2001 to 2016.

The extremely impacted periphery areas of wetlands and river bank areas of the lower Meghna floodplain were conversely correlated of LST and SINDVI (Figure 6a,b). In the extremely changed wetlands area, the SINDVI was increased by 3.1 indexed values for an 0.8 °C/year increase of temperature. The river bank area of the lower Meghna floodplain had an increase in SINDVI of 5.0 indexed values for 2.9 °C/year increased LST.

The extreme impacts on some grid cells of southern tidal floodplains, and inner and outer portion of the coastal plains were inversely correlated with LST and SINDVI (Figure 8c,d,f). The interannual average SINDVI was increased by 5.9 indexed values in the extremely changed grid cell of the tidal floodplain region and had 1.3 °C/year decreased temperatures. While 1.9 °C/year decreased temperatures were increased, 1.8 indexed SINDVI increased at the inner and inundation free coastal plain region. The outer coastal plain region as well as occasionally inundated grid cells had losses of vegetation of 1.7 indexed values for 2.3 °C/year temperature increases. The LST and SINDVI were slightly negatively correlated in the hilly areas with the most impacted grid cells as well as in the forest covered region of the study (Figure 8e).

### 4.2. Geospatial Relation Anomalies

The interannual average LSTand SINDVI in the period of study from 2001 to 2016 demonstrated a diverse relationship with their geospatial locations (Figure 9). The yearly average values of LST and SINDVI in the regions of wetlands, estuarine floodplains, middle and lower Meghna River floodplains, and terraced lands were conversely interrelated with different trends (Figure 9a–d,g). For all of those land-type areas, the LSTs and SINDVI increased simultaneously.

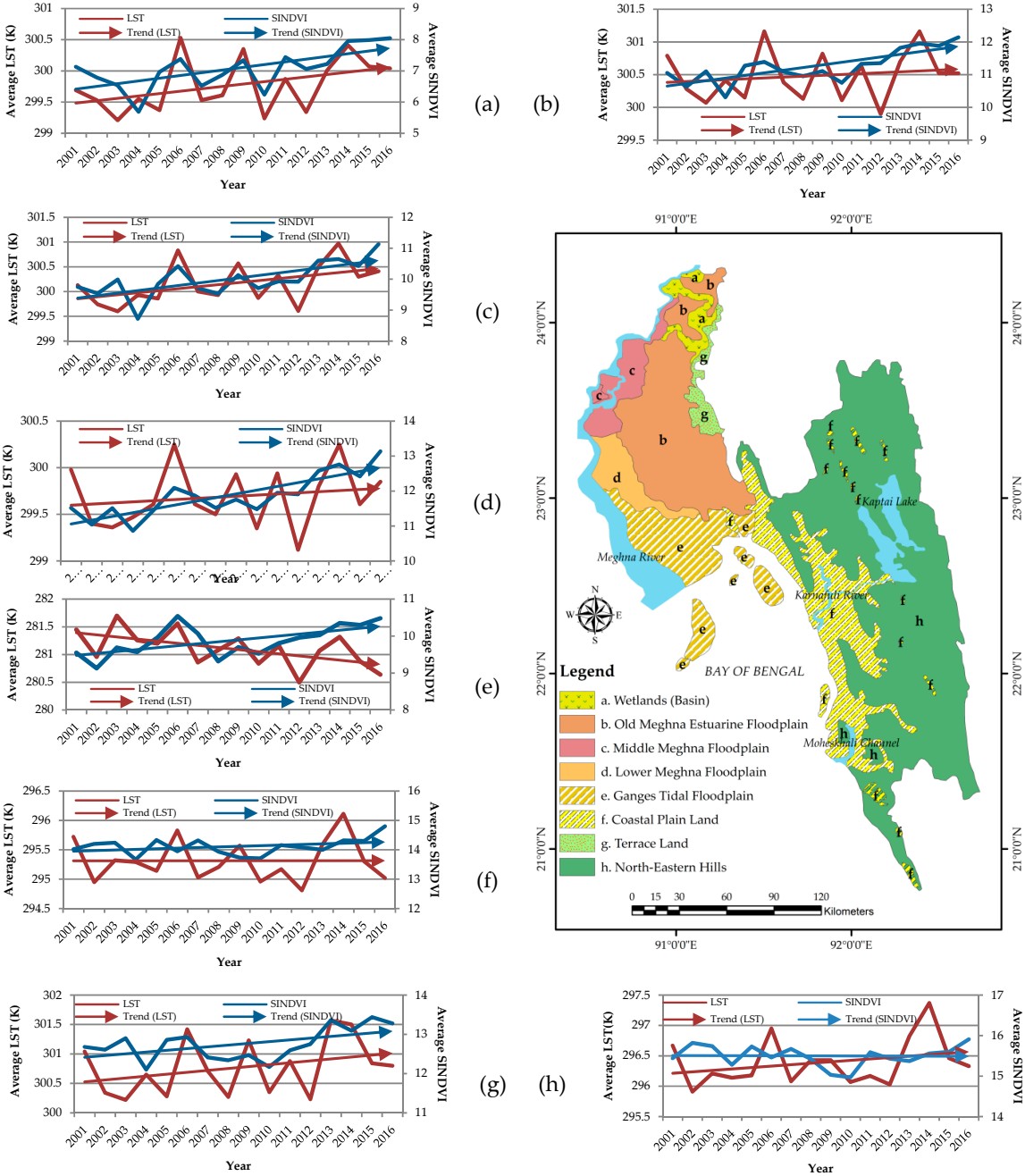

**Figure 9.** Interannual averages of LST and SINDVI in (**a**) wetlands, (**b**) old Meghna estuarine floodplain, (**c**) middle Meghna River floodplain, (**d**) lower Meghna River floodplain, (**e**) Ganges tidal floodplain, (**f**) coastal plain lands, (**g**) terraced lands, and (**h**) Northeastern Hills at the southeastern region of Bangladesh.

In the areas of tidal floodplains, coastal plain lands, and Northeastern Hills, the interannual relations were quite inversed (Figure 9e,f,h). The decreasing years of LST at the tidal floodplain areas have increased the SINDVI in high proportions. Although the land surface temperatures were mostly static at the coastal plain lands, the yearly average SINDVIs have fluctuated and increased. The hilly and mountain lands vegetation dynamism was mostly static with a little fluctuating in the interannual index, although it showed increasing temperatures.

The entire indexes of the LST were found to have increasing trends overall land types except tidal floodplains and coastal plain lands in the southeastern region of Bangladesh during the period of study (Figure 10). These increased temperatures acted as mostly positive consequences on vegetation dynamics for the plain land areas as well negative for hilly and mountainous region.

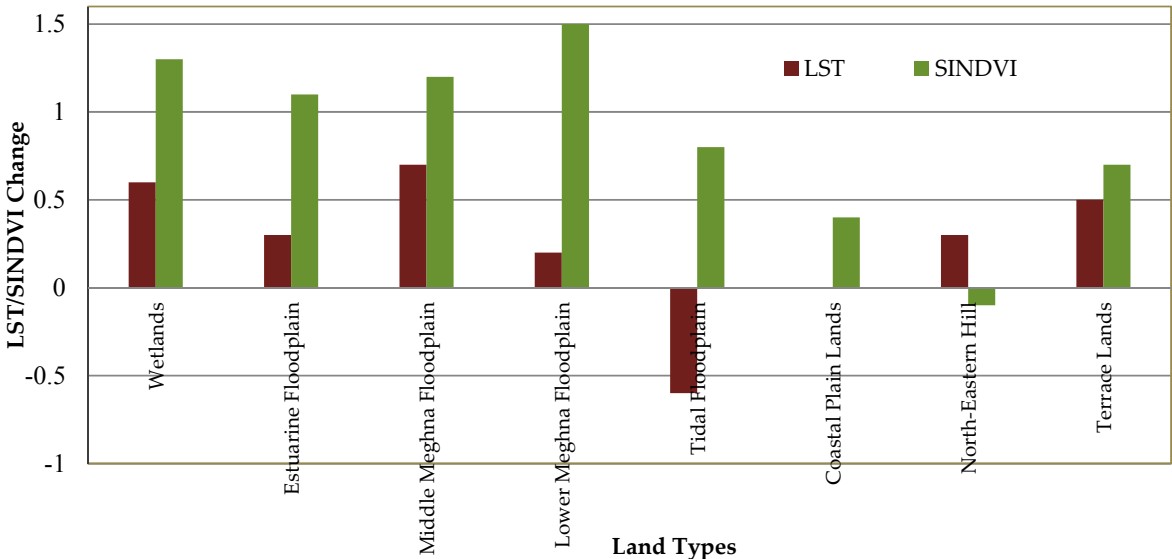

**Figure 10.** Geospatial relations of LST and SINDVI range by mean value average.

The middle Meghna River floodplain region experienced the highest increase in temperature (0.6 °C/16years) and a moderate SINDVI increase (1.3 indexed). The highest vegetation increase (1.5 indexed) was in the lower Meghna floodplain with only 0.2 °C/16 years increase in temperatures. The hilly and mountain regions lost 0.1 indexed value of SINDVI for 0.3 °C/16years temperature increase.

The decreasing and mostly static temperature changes in the study areas acted to increase vegetation dynamics. The tidal floodplain region received a 0.8 indexed value of SINDVI for 0.6 °C/16years temperature decrease, and coastal plain land received 0.4 indexed value of SINDVI for its stable temperatures.

### 4.3. Limitations and Future Research

In this study, the land surface temperature and vegetation index were the factors considered and estimated for monitoring the trends in the LST and SINDVI. However, other parameters such as air temperature, albedo, precipitation, tree cover, and soil moisture [31,44] can also be important for the relationship of LST and vegetation dynamics. Unfortunately, due to the limitation of reliable high-resolution data on the regional level as well as study area, the changes of these parameters have not been studied in the current research. However, we recognize the potential importance of other parameters. In future studies, including more parameters will clearly increase our understanding of the relationship of LST and vegetation dynamics.

## 5. Conclusions

Within the context of climate change and accelerating rates of regional development, an understanding of the manner in which the LST and SINDVI might offset consequential negative environmental and social impacts becomes imperative. This study focused on the geospatial relationship analysis of land surface temperatures on vegetation dynamics in the southeastern region of Bangladesh. As a vulnerable and disaster-prone area due to climate change, most of the research focused here pertains to temperature change, especially of air temperatures [10,12–15]. Recognition and awareness of this functionality in the relationship of LST and SINDVI is critical as well as crucial for development. It can serve as a tool for adapting to climate change and reducing many of the negative impacts of development through improved planning and management strategies. This study found that within the southeastern region of Bangladesh, LST and SINDVI shared a significant inverse and converse relationship according to their areas of impact. While LSTs were increasing in the study areas, the hilly regions decreased their vegetation and vast areas of floodplain regions expanded. However, this relationship has distinct differences depending on land-use and land-cover (LULC) types. Increasing the amount of vegetation in some LULC types will not be as effective at lowering temperatures as in others. It is hoped that the findings presented in this study may prove useful to those involved in climate change, environmental and agricultural study and ecological planning in this region as well as Bangladesh, in terms of illuminating which types of actions may be the most beneficial.

**Author Contributions:** S.I. and M.M. conceived and designed the research; S.I. performed the experiments and analyzed the data; S.I. and M.M. jointly revised the paper.

**Funding:** This work was jointly supported by the National Natural Science Foundation of China (grant number: 41771453 & 41601448), Special Project of Science and Technology Basic Work (grant number: 2014FY210800-5) and Chongqing R&D Project of the high technology and major industries [2017] 1231.

**Acknowledgments:** In this study, multi-resource data were downloaded from different data centers. The data include MODIS LST and NDVI data obtained from Atmospheric Composition Analysis Group. The authors express their gratitude for the data sharing of above datasets. The authors would also like to thank Yu Wenping and Ni Xiang for the modification of the manuscript. We sincerely appreciate the three anonymous reviewers' constructive comments and the editor's efforts in improving this manuscript.

**Conflicts of Interest:** The authors declare no conflict of interest.

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
