# Peer review of "Geospatial Monitoring of Land Surface Temperature Effects on Vegetation Dynamics in the Southeastern Region of Bangladesh from 2001 to 2016"

_ijgi, doi:10.3390/ijgi7120486_

Round 1

Reviewer 1 Report

The study aim is to present the Research on “Geospatial monitoring of Land Surface Temperature on Vegetation dynamics at the Southeastern Region of Bangladesh from 2001 to 2016” The manuscript is presented in a clear and nice way. However to have scientific merit the following comments should be addressed.

1.      The main purpose of the paper is not clear in the introduction.

2.      The image usage paper is not clear. How many images used to identify the LST and SINDVI trend? Better to add details

3.      Figure 3: the LST and SINDVI better to classify into several class

4.      The procedure used to create Figure 5 is not clear. How did you incorporate LST and SINDVI into one map

5.      Figure 6: how did you select these six locations and why these six locations different from other areas. Need to explain

6.      Better to combine LST and SINDVI in one graph with two Y - axis

7.      The locations of Figure 7 need to show in the maps and better to shows the Average LST and Average SINDVI in one graph. You can use two Y-axis. The scatterplot plot can be used to plot two variables.

8.      Discussion and conclusion need to improve with more focusing on the finds of the this research

9.      All figures neet to convert high resolution

Author Response

The study aim is to present the Research on “Geospatial monitoring of Land Surface Temperature on Vegetation dynamics at the Southeastern Region of Bangladesh from 2001 to 2016” The manuscript is presented in a clear and nice way. However to have scientific merit the following comments should be addressed.

Response: Thank you for your very positive comments on our manuscript, which are greatly appreciated. After serious consideration, we have adopted virtually all of the key suggestions. Based on your suggestions, we have checked and modified the manuscript. Following the revisions, we believe that the quality of the manuscript has been improved considerably.

Point 1: The main purpose of the paper is not clear in the introduction.

Response 1: Thank you for your observation. We have re-write and modified the introduction section according to your valuable suggestions. Moreover, we have added the innovation point and the main purpose of this paper as you mentioned (L88-95).

Point 2: The image usage paper is not clear. How many images used to identify the LST and SINDVI trend? Better to add details

Response 2: Thank you for your reminder and quarry. For LST trend analysis, we have used MODIS LST products of 8 days cumulative differences of total 736 images and 368 images for SINDVI trend analysis by 16 days cumulative differences MODIS NDVI products. We have added in detail L133-137 and L155-157.

Point 3: Figure 3: the LST and SINDVI better to classify into several class

Response 3: Thank you for your constructive suggestions. We have classified the variation trends of LST and SINDVI into several classes to represent the change as well (Figure 4a and 4b).

Point 4: The procedure used to create Figure 5 is not clear. How did you incorporate LST and SINDVI into one map?

Response 4: Thank you for your quarry. We have calculated the spatial patterns of correlation coefficient of LST and SINDVI by their yearly average and total time period (16 years) average value of analysis (Now it is in Figure 7a). The Pearson’s correlation coefficients were used to determine the correlations on each pixel. We have also added a significant test by critical values of Pearson’s correlation (Figure 7b). In accordance to your quarry, we have included the expression in L304-306 and added description L322-328.

Point 5: Figure 6: how did you select these six locations and why these six locations different from other areas. Need to explain

Response 4: Thank you for your asking and valuable suggestion. The six points in Figure 6 (Now Figure 8), we have mainly selected by their extremely increased (>4℃) and decreased (˂-4℃) temperature, extremely increased (>10.0) and decreased (˂-10.0) SINDVI index, and as well as no changes of the trends of LST and SINDVI. It was also tried to keep the distributions all over the study areas. The focuses were mainly on relationship of LST and SINDVI at the extremely impacted areas. The expression has cleared in L348-354.

Point 6: Better to combine LST and SINDVI in one graph with two Y – axis

Response 6: Thank you for your constructive suggestion. We have re-arranged the figure of LST and SINDVI in combined both elements into one respective graph with two Y –axis (now in Figure 8).

Point 7: The locations of Figure 7 need to show in the maps and better to shows the Average LST and Average SINDVI in one graph. You can use two Y-axis. The scatter plot can be used to plot two variables.

Response 7: Thank you for your reproductive suggestions. We have inserted the physiographic land type’s map in the Figure with locations based on analysis (Now in Figure 9).

In addition, the figures of average LST and average SINDVI have re-arranged in combined both elements into one respective graph with two Y –axis (Figure 9).

Thank you for your suggestion about scatter plot. In respect upon you, we think that the line diagram will more perfectly represent the results because we want to show the variable in continuous distribution.

Point 8: Discussion and conclusion need to improve with more focusing on the finds of this research.

Response 8: Thank you for your productive suggestions. We have re-write and modified the discussion and conclusion section with some more focusing on the findings of this research.

Point 9: All figures need to convert high resolution.

Response 9: Thank you for your reminder and suggestions. All figures in this paper we have converted and modified with high resolution.

Reviewer 2 Report

This paper assesses geospatial patterns of LST in correlation of SINDVI at the southeastern Bangladesh from 2001 to 2016. In general, the paper is easy for the reader to follow. Although the topic is interesting, I do not think this paper should be published in IJGI with present version. The major problems can be summarized as follows: 

1.       The introduction section should be re-written. I would suggest the authors to reduce descriptions of study area. In fact, I think one paragraph is enough. The research on climate change impact of Bangladesh is actually centered on air temperature. However, this study used land surface temperature. These two parameters are different. The authors should carefully think about that. Besides, the authors should put some efforts on the previous literature researching on the relationship between SINDVI and LST. What is the gap in previous studies that this study tries to fill? The authors should explicitly identify that. Moreover, in the last paragraph, the innovation point and aim of this paper should be explicit.

2.       Why the authors use SINDVI instead of NDVI, the latter of which is more widely used in previous studies? Please provide explanations or justifications for that.

3.       Too much descriptions for study area. I would suggest the authors to condense that.

4.       In section 2.2, the reference 22 seems to be irrelevant. It analyzed radiative forcing due to albedo change caused by land cover change in China, rather than evaluating MODIS LST and NDVI products. In addition, the authors mentioned 30m landsat tree cover products. But actually, I didn’t see the authors use this product throughout the paper.

5.       Did the authors check the significance of the trends as well as correlations?

6.       In the Discussion section, the author should add some previous references and relevant findings related to your study to give more insights.

7.       In the Conclusion section, what is the limitation of this study? I would suggest the authors to add on that as well as the future work of this study.

Author Response

This paper assesses geospatial patterns of LST in correlation of SINDVI at the southeastern Bangladesh from 2001 to 2016. In general, the paper is easy for the reader to follow. Although the topic is interesting, I do not think this paper should be published in IJGI with present version.

Response: Thank you for your very positive comments on our manuscript, which are greatly appreciated. After serious consideration, we have adopted virtually all of the key suggestions. Based on your suggestions, we have checked and modified the manuscript. Following the revisions, we believe that the quality of the manuscript has been improved considerably.

The major problems can be summarized as follows:

Point 1: The introduction section should be re-written. I would suggest the authors to reduce descriptions of study area. In fact, I think one paragraph is enough. The research on climate change impact of Bangladesh is actually centered on air temperature. However, this study used land surface temperature. These two parameters are different. The authors should carefully think about that. Besides, the authors should put some efforts on the previous literature researching on the relationship between SINDVI and LST. What is the gap in previous studies that this study tries to fill? The authors should explicitly identify that. Moreover, in the last paragraph, the innovation point and aim of this paper should be explicit.

Response 1: Thank you for your constructive suggestions.  We have also realized the matter that many research on vegetation especially forest cover change were identified at the study area in relation to air temperature. While, there have limited research based on LST as we found. Although, we have identified so many research paper related to these variable in globally. Therefore, in lacking of LST based research of the study area, we have presented some literature review based on the air temperature changes, in fact it also have direct or indirect impacts/relation with land surface temperature.

In based on your reproductive suggestions, we have re-write and modified the introduction section. The modifications tried represent the previous study gap and how this study may overcome. Moreover, we have added the innovation point and aim of this research as you mentioned. We have changed some areas expression of introduction in L59-62 and L88-95 as well.

Response 3 has described about the modification 'study area'.

Point 2: Why the authors use SINDVI instead of NDVI, the latter of which is more widely used in previous studies? Please provide explanations or justifications for that.

Response 2: Thank you for your asking and productive suggestions. As we know, the seasonally integrated normalized difference vegetation index (SINDVI) is defined as the sum of NDVI values for each pixel and all time intervals of maximum value composites (MVCs) (Holben 1986) that the NDVI exceeds a critical value (commonly NDVI>0.1). NDVI has been using by many research work, is the most widely used vegetation index.

However, there are frequent fluctuations with remarkable rises and falls because of the variation of cloudiness, data transmission errors, and incomplete or inconsistent atmospheric correction, and bi-directional effects in the NDVI magnitudes. Vegetation constitutes the natural interface among air, water and soil. It typically shows seasonal and annual dynamics where use SINDVI is preferable (Stow, 2004).

Another important issue when analyzing vegetation changes from NDVI data sets is the effect that stratospheric aerosols (SAs) have on NDVI magnitudes. In an attempt to minimize the SA effect and emphasize trends in vegetation greenness, SINDVI is successful (Stow, 2001).

A generalized NDVI (SINDVI) temporal profile is continuous and smooth because vegetation canopy changes are small with respect to time (Stow, 2003).

In addition, the study area is mostly covered by Indo-Burma biodiversity hotspot with mixed up of multi types of vegetation cover and very little seasonal fluctuations. So, we have used the yearly variations instead of seasons.

References:

Holben, B. Characteristics of maximum value composites images for temporal AVHRR. International Journal of Remote Sensing, 1986, 7, 165-174.

Stow, D.; Daeschner, s.; Hope, A.; Douglas, D.; Mynane, R.; Zhou, L. Spatial-Temporal Trend of Seasonally-integrated Normalized Difference Vegetation Index as an Indicator of Changes in Arctic Tundra Vegetation in the Early 1990s. Proceedings of the International Geosciences and Remote Sensing Symposium, Sydney, Australia. 2001. IEEE, 1, 181-183.

Stow, D.; Daeschner, S.; Hope, A.; Douglas, D.; Petersen, A.; Myneni, R.; Oechel, W. Variability of the seasonally integrated normalized difference vegetation index across the north slope of Alaska in the 1990s. International Journal of Remote Sensing, 2003, 24(5), 1111–1117, https://doi.org/10.1080/0143116021000020144.

Stow, D.A., Hope, A., McGuire, D., et al. Remote sensing of vegetation and land-cover change in Arctic Tundra Ecosystems. Rem. Sens. Environ. 2004, 89, 281–308.

Point 3: Too much descriptions for study area. I would suggest the authors to condense that.

Response 3: Thank you for your constructive suggestion. We have modified and condensed the descriptions of the study area according to your suggestion. However, we have continued the second paragraph in as usual of physiographic types as we described inter-annual changes of the variables.

Point 4: In section 2.2, the reference 22 seems to be irrelevant. It analyzed radiative forcing due to albedo change caused by land cover change in China, rather than evaluating MODIS LST and NDVI products. In addition, the authors mentioned 30m landsat tree cover products. But actually, I didn’t see the authors use this product throughout the paper.

Response 4: Thank you for your comments and reminder. Actually, we have just taken the regionally MODIS time series data evaluation procedure from the reference 22. As it is irrelevant, later we can remove it. We are very sorry for our negligence of irrelevant line "30m landsat tree cover products" and already removed this line in revised paper.

Point 5: Did the authors check the significance of the trends as well as correlations?

Response 5: Thank you for your asking and reminder. In accordance to your suggestion, we have added the conventional non-parametric Mann-Kendall test (M-K test) and has been extensively used to assess the significance of monotonic trends in LST and SINDVI time series [43, 44, 45] (L179-181) and the results have shown (Figure 5) and described at the results section under 3.1 Variation Trends of LST and SINDVI (L270-285).

In addition, for correlation calculation, Student’s t test was applied to assess statistical significance and table of critical values of Pearson correlation was used to significance distribution (L189-190). The results have shown (Figure 7b) and described in section 3.3 Correlation with LST and SINDVI (L320-328).

Point 6: In the Discussion section, the author should add some previous references and relevant findings related to your study to give more insights.

Response 6: Thank you for your constructive suggestions. We have modified the Discussion section with some previous references and relevant findings in accordance to you suggestions.

Point 7: In the Conclusion section, what is the limitation of this study? I would suggest the authors to add on that as well as the future work of this study.

Response 7: Thank you for your constructive suggestion. We have included the limitation and future research by a separated section as 4.3 Limitations and Future research under discussion section. Now it is distributed in L413-421.

Reviewer 3 Report

General Remarks

I believe this manuscript is appropriate to be published within the ISPRS International Journal of Geo-Information since it contains significant geo-information content. The paper addresses the geospatial relationship between LST and NDVI and explores inter-annual trends of these variables over the period of 2001 – 2016, for south-eastern region of Bangladesh. Although it does not bring any big innovation to the scientific community this work may be an important asset for local applications in Bangladesh. For this reason, I believe the paper is suited for future publication. However, it must not be accepted as it stands. Several improvements need to be made before publishing the paper. I recommend major revisions.

Below I enumerated some general remarks that the authors must tackle to improve the manuscript.

There are several examples of poor writing English. Below I show some of them, but there are many more that need to be tackled.

The sentence:

“In general, 63.41% of the study areas were found increased LST by the year 2001 – 2016 (Figure 2a).”

Is very difficult to understand, and may be improved by altering to the following:

“Overall, about 63% of the total area of study shows an increase in LST in the period 2001 – 2016 (Figure 2a).”

Another example is the sentence:

The inter-annual average SINDVI increased in some areas increasing and some others decreased.”

Which does not bring anything new to the paper and does not make any sense.

Since lines are not enumerated it is very hard for me to point you to other errors that I found. I urge the authors to carefully review the written English of the paper. I understand that English may not be the native language of the authors. However, in a case like this I recommend asking for help from some English-speaking colleague or from some company dedicated to that kind of work. It is crucial that the authors change the text, as it cannot be accepted as it is.

Specific Remarks

1.       Section 2.2.1 (LST section)

How did the authors create the yearly composites of LST? What is the impact of the number of clear-sky observations on the final composite. The authors should show a map of the total number of observations per pixel, or at least mention it in the text.

2.       Section 2.2.2 (NDVI section)

The reason why the authors use a seasonal integrated NDVI (SINDVI) is not clear for me. Why not use NDVI as it is usually taken? If there is a reason for that the authors must explain it in a detailed form, since this approach is not the most common. Furthermore, the authors use this SI NDVI but then do not divide the year in seasons (see point 8.).

3.       The sections 2.2, 2.2.1, 2.2.2 are repeated… please rectify.

4.       Section 2.2.1 (Time Series Analysis)

The authors refer that if the slope is near 0 there are no significant changes. This is a poor approach. The authors must include a statistical significance test, e.g. the Mann-Kendall test, to verify the significance of the trends.

5.       Section 2.2.2 (Correlation and…)

Once again, correlations need to be statistically tested with a parametric or non-parametric test.

6.       I also believe that equations must be numbered… Also, the first and third equations are unnecessary. However, if the authors want to keep them, they should replace X and Y for the respective variable.

7.       Section 2.3

Define MVI. Respective reference not in the journal’s format.

The authors also need to better explain why they used this reconstructed NDVI dataset. It is not clear. There are too many statistical tools applied to NDVI (first the SINDVI, then the reconstructed). This is a very confusing way, that need to have a very good reason to be done. The reason is not clearly described in the paper. Please, reformulate the methods sections to be clearer.

8.       Section 3 and 4

What is the meaning of “the proportion of LST …”? Why not use the usual units of trend for a temperature (ºC/year; ºC/decade, etc)?

One of my main concerns in this paper is relative to why the results are shown for annual means. It would be much more interesting if the results were divided in seasons. I am not acquainted of the region, but I imagine that at least there are 2 seasons (winter, summer) to study. I am not sure if spring and autumn are of interest for the region. Results could have been much more interesting if the authors made this intra-annual division and then for the different seasons they performed the inter-annual analysis. Since the paper revolves around the spatio-temporal variability of LST and NDVI, the authors must try to further understand the details of such variability.

The authors also need to better discuss the results. Several papers addressed the problem of greening and browning of vegetation activity. These results must be taken in consideration. You are studying LST and NDIV (or SINDVI…), which are two of the most used remote sensing variables to address drought monitoring. Many studies have taken in consideration the relation between these two variables and its effect over vegetation health (most notably the Vegetation Health Index – VHI). On these studies it is several times shown that the positive or negative relation between LST and NDVI is seasonal, with a positive relation in winter and a negative in summer; and that the relation may also depend on the land cover. The authors must complete the discussion with this information and explain what novelty this analysis brings, linking it to (e.g.) the agriculture of the region.

Finally, the quality of the maps must be improved.

I strongly recommend the authors to reformulate the methods section and improve the quality of the results and also improve discussion. For these reasons I recommend major revisions. The authors may choose to later resubmit in order to have more time to tackle these issues.

Author Response

General Remarks:

I believe this manuscript is appropriate to be published within the ISPRS International Journal of Geo-Information since it contains significant geo-information content. The paper addresses the geospatial relationship between LST and NDVI and explores inter-annual trends of these variables over the period of 2001 – 2016, for south-eastern region of Bangladesh. Although it does not bring any big innovation to the scientific community this work may be an important asset for local applications in Bangladesh. For this reason, I believe the paper is suited for future publication. However, it must not be accepted as it stands. Several improvements need to be made before publishing the paper. I recommend major revisions.

Below I enumerated some general remarks that the authors must tackle to improve the manuscript.

There are several examples of poor writing English. Below I show some of them, but there are many more that need to be tackled.

The sentence:

“In general, 63.41% of the study areas were found increased LST by the year 2001 – 2016 (Figure 2a).”

Is very difficult to understand, and may be improved by altering to the following:

“Overall, about 63% of the total area of study shows an increase in LST in the period 2001 – 2016 (Figure 2a)”.

Another example is the sentence:

The inter-annual average SINDVI increased in some areas increasing and some others decreased.”

Which does not bring anything new to the paper and does not make any sense.

Since lines are not enumerated it is very hard for me to point you to other errors that I found. I urge the authors to carefully review the written English of the paper. I understand that English may not be the native language of the authors. However, in a case like this I recommend asking for help from some English-speaking colleague or from some company dedicated to that kind of work. It is crucial that the authors change the text, as it cannot be accepted as it is.

Response: Thank you for your positive comments on our manuscript, which are greatly appreciated. After serious consideration, we have adopted virtually all of the key suggestions in your general and specific remarks. Based on your suggestions, we have checked and modified the manuscript. Following the revisions, we believe that the quality of the manuscript has been improved considerably.

Specific Remarks:

Point 1: Section 2.2.1 (LST section)

How did the authors create the yearly composites of LST? What is the impact of the number of clear-sky observations on the final composite? The authors should show a map of the total number of observations per pixel, or at least mention it in the text.

Response 1: Thank you for your quarry and constructive suggestions. We have created the each year composites of LST over yearly in averaged of 46 MODIS LST products of Eight-day compositing period. A simple average method is used in the current algorithm on each pixel and all time intervals of maximum value composites (MVCs) [41, 42]. Now it is re-arranged and added at L133-137.

Thanks for asking about the impact number of clear-sky observations. We are very sorry for our negligence. In according to your suggestions, we have calculated the number of clear-sky day and clear-sky night on each pixel for one year composites of LST of 2016. In accordance to your suggestion, we have included two maps as the total number of observations of clear-sky day and clear-sky night of 2016 added in Figure 2a and 2b in the revised paper (L138-143).

Point 2: Section 2.2.2 (NDVI section)

The reason why the authors use a seasonal integrated NDVI (SINDVI) is not clear for me. Why not use NDVI as it is usually taken? If there is a reason for that the authors must explain it in a detailed form, since this approach is not the most common. Furthermore, the authors use this SI NDVI but then do not divide the year in seasons (see point 8.).

Response 2: Thank you for your asking and productive suggestions. As we know, the seasonally integrated normalized difference vegetation index (SINDVI) is defined as the sum of NDVI values for each pixel and all time intervals of maximum value composites (MVCs) (Holben 1986) that the NDVI exceeds a critical value (commonly NDVI>0.1). NDVI has been using by many research work, is the most widely used vegetation index.

However, there are frequent fluctuations with remarkable rises and falls because of the variation of cloudiness, data transmission errors, and incomplete or inconsistent atmospheric correction, and bi-directional effects in the NDVI magnitudes. Vegetation constitutes the natural interface among air, water and soil. It typically shows seasonal and annual dynamics where use SINDVI is preferable (Stow, 2004).

Another important issue when analyzing vegetation changes from NDVI data sets is the effect that stratospheric aerosols (SAs) have on NDVI magnitudes. In an attempt to minimize the SA effect and emphasize trends in vegetation greenness, SINDVI is successful (Stow, 2001).

A generalized NDVI (SINDVI) temporal profile is continuous and smooth because vegetation canopy changes are small with respect to time (Stow, 2003).

In addition, the study area is mostly covered by Indo-Burma biodiversity hotspot with mixed up of multi types of vegetation cover and very little seasonal fluctuations. So, we have used the yearly variations instead of seasons.

References:

Holben, B. Characteristics of maximum value composites images for temporal AVHRR. International Journal of Remote Sensing, 1986, 7, 165-174.

Stow, D.; Daeschner, s.; Hope, A.; Douglas, D.; Mynane, R.; Zhou, L. Spatial-Temporal Trend of Seasonally-integrated Normalized Difference Vegetation Index as an Indicator of Changes in Arctic Tundra Vegetation in the Early 1990s. Proceedings of the International Geosciences and Remote Sensing Symposium, Sydney, Australia. 2001. IEEE, 1, 181-183.

Stow, D.; Daeschner, S.; Hope, A.; Douglas, D.; Petersen, A.; Myneni, R.; Oechel, W. Variability of the seasonally integrated normalized difference vegetation index across the north slope of Alaska in the 1990s. International Journal of Remote Sensing, 2003, 24(5), 1111–1117, https://doi.org/10.1080/0143116021000020144.

Stow, D.A., Hope, A., McGuire, D., et al. Remote sensing of vegetation and land-cover change in Arctic Tundra Ecosystems. Rem. Sens. Environ. 2004, 89, 281–308.

Point 3: The sections 2.2, 2.2.1, 2.2.2 are repeated… please rectify

Response 3: Thank you for your reminder. We are very sorry for our negligence of the repeat numbering. The next repeated number have rectified as well.

Point 4: Section 2.2.1 (Time Series Analysis)

The authors refer that if the slope is near 0 there are no significant changes. This is a poor approach. The authors must include a statistical significance test, e.g. the Mann-Kendall test, to verify the significance of the trends.

Response 4: Thank you for your constructive suggestion and analysis. In accordance to your suggestion, we have added the conventional non-parametric Mann-Kendall test (M-K test) and has been extensively used to assess the significance of monotonic trends in LST and SINDVI time series [43, 44, 45] (L179-181) and the results have shown (Figure 5) and described at the results section under 3.1 Variation Trends of LST and SINDVI (L270-285).

Point 5: Section 2.2.2 (Correlation and…)

Once again, correlations need to be statistically tested with a parametric or non-parametric test.

Response 5: Thank you for your constructive suggestion. In addition, for correlation calculation, Student’s t test was applied to assess statistical significance and table of critical values of Pearson correlation was used to significance distribution (L189-190). The results have shown (Figure 7b) and described in section 3.3 Correlation with LST and SINDVI (L320-328).

Point 6: I also believe that equations must be numbered… Also, the first and third equations are unnecessary. However, if the authors want to keep them, they should replace X and Y for the respective variable

Response 6: Thank you for your constructive suggestions. We have re-arranged the equations with numbered as well 1, 2, 3,…

In respect of your suggestions, we have reorganized it. Actually, it was used the slope trends of LST and SINDVI according to equation 1 and Pearson’s correlation coefficient by equation 3. Expectantly, it will be better to keep these equations for the description of ‘Time series analysis’ section.

In respect to your suggestion about replace X and Y, we have rethought the matter. In fact, the same equation (equation 1) was used for the changing rates of LST and SINDVI slope calculations and a common symbol ‘A’ was used as well. Although, we have already used X and Y for the respective variables in equation 3.

Point 7: Section 2.3

Define MVI. Respective reference not in the journal’s format.

The authors also need to better explain why they used this reconstructed NDVI dataset. It is not clear. There are too many statistical tools applied to NDVI (first the SINDVI, then the reconstructed). This is a very confusing way that needs to have a very good reason to be done. The reason is not clearly described in the paper. Please, reformulate the methods sections to be clearer.

Response 7: Thank you for your quarry and suggestions. In accordance to your quarry, the mean-value iteration filter (MVI) is a simple but very effective reconstruction method, has been developed to reduce the noise and to enable the reconstruction of high quality NDVI time-series [29, 35, 38, 39].

We are agreed to you that there are so many statistical tools to reconstruct NDVI. Other than we have used this method as for the maximum value composite (MVC) applied on the NDVI is commonly used to reduce the error sources [41] and the cloud detection identification algorithm (CLAVR) is often used (Stow et al., 1991), significant residual effects remain. Therefore, some scientists explored a number of methods to reduce the noise levels and to reconstruct high quality NDVI time-series data sets, like for example the best index slope extraction (BISE) method (Viovy et al., 1992), the modified BISE filtering (Lovell and Graetz, 2001), and a fast Fourier transform (FFT) algorithm (Verhoef et al., 1996), which are both widely used in reducing noise of NDVI time series.

The BISE algorithm was designed for the daily NDVI time series (Viovy et al., 1992). This algorithm cannot be as effective with our datasets of the 16-day composite data.

Adjustments had been made to the BISE rules for Australia with the PAL NDVI data (Lovell and Graetz, 2001). The spurious high or low points were modified to look for a spike.

The FFT algorithm can make the most smoothed profile of NDVI time series, but all of the points show large displacement away from the original points, which can be shown in all the sample pixels. The symmetric sine and cosine functions of the FFT calculate the noise NDVI values and obtain an intermediate value, which not only adjust the spurious points but also confuse the right points.

In contrast with the FFT algorithm, the modified BISE algorithm and MVI method keep the original value of most of the points and just adjust the remarkable abnormal fluctuations. It can be seen that most of the noisy points were successfully identified and corrected by using these two methods. But the modified BISE-reconstructed NDVI time series cannot remove some of the higher value points.

It can be indicated that although the new MVI filter is very simple, it is quite effective in reconstructing temporally and spatially more homogeneous PAL NDVI time series.

The choice of the threshold value is important in the newly developed method. Most of points would be retained if the threshold value were too high. On the contrary, most of points would be adjusted if the threshold value were too low, which was just same as the problems the FFT filter.

Due to the quality improvement, essentially elicited as noise reduction. The improved NDVI time series can therefore be applied more effectively in the monitoring of inter-annual vegetation changes. Clearly an improved NDVI time series leads to a retrieval of biophysical land surface variables with higher quality.

Point 8: Section 3 and 4

What is the meaning of “the proportion of LST …”? Why not use the usual units of trend for a temperature (ºC/year; ºC/decade, etc)?

One of my main concerns in this paper is relative to why the results are shown for annual means. It would be much more interesting if the results were divided in seasons. I am not acquainted of the region, but I imagine that at least there are 2 seasons (winter, summer) to study. I am not sure if spring and autumn are of interest for the region. Results could have been much more interesting if the authors made this intra-annual division and then for the different seasons they performed the inter-annual analysis. Since the paper revolves around the spatio-temporal variability of LST and NDVI, the authors must try to further understand the details of such variability.

The authors also need to better discuss the results. Several papers addressed the problem of greening and browning of vegetation activity. These results must be taken in consideration. You are studying LST and NDIV (or SINDVI…), which are two of the most used remote sensing variables to address drought monitoring. Many studies have taken in consideration the relation between these two variables and its effect over vegetation health (most notably the Vegetation Health Index – VHI). On these studies it is several times shown that the positive or negative relation between LST and NDVI is seasonal, with a positive relation in winter and a negative in summer; and that the relation may also depend on the land cover. The authors must complete the discussion with this information and explain what novelty this analysis brings, linking it to (e.g.) the agriculture of the region.

Finally, the quality of the maps must be improved

I strongly recommend the authors to reformulate the methods section and improve the quality of the results and also improve discussion. For these reasons I recommend major revisions. The authors may choose to later resubmit in order to have more time to tackle these issues

Response 8: Thank you very much for your quarry and elementary suggestions. We have described the total changes of temperature in the time period as proportion of LST. The changes of LST will be 0C/year as we analyzed as per year. We are very sorry for our mistake and thank you for your reminder.

In respect of your suggestions, the vegetation dynamics were used as annual because a generalized NDVI (SINDVI) temporal profile is continuous and smooth because vegetation canopy changes are small with respect to time [23]. Moreover, the study area is mostly covered by Indo-Burma biodiversity hotspot with mixed up of multi types of vegetation cover and very little seasonal fluctuations [18]. So, we have used the yearly variations instead of seasons.

Thank you for your constructive suggestions about the improvement of discussion. We have tried to improved discussion section.

Thanks for the suggestion to improve the quality of maps. We have changed all the maps and related graphs with high resolution and recommended changes.

For you kind regards, we have done possible reformulated in the methods section, improved results and discussion according to your valuable and productive suggestions.

Round 2

Reviewer 1 Report

I am happy with the revised manuscript and the corrections which I asked in the previous version.

Author Response

Comments and Suggestions: I am happy with the revised manuscript and the corrections which I asked in the previous version.

Response: Thank you for your very positive comments on our revised manuscript and accepting our revision. We are greatly appreciated by your reproductive suggestions.

In addition, we have edited our revised manuscript for proper English language, grammar, punctuation, spelling, and overall style by the editors related to this research field at “American Journal Experts (AJE)” (Editorial Certificate is given as attachment). Following the revisions, we believe that the quality of the manuscript has been improved considerably.

Reviewer 3 Report

The authors answered to all my questions in an acceptable fashion and thus it is now in acceptable form to publish.

Nevertheless, the written English still needs to be checked.

Author Response

Comments and Suggestions: The authors answered to all my questions in an acceptable fashion and thus it is now in acceptable form to publish.

Nevertheless, the written English still needs to be checked.

Response: Thank you for your very positive comments on our revised manuscript and accepting our revision. We are greatly appreciated by your reproductive suggestions.

Based on your suggestion, we have edited our revised manuscript for proper English language, grammar, punctuation, spelling, and overall style by the editors related to this research field at “American Journal Experts (AJE)” (Editorial Certificate is given as attachment). Following the revisions, we believe that the quality of the manuscript has been improved considerably.